# Current Insights in Elucidation of Possible Molecular Mechanisms of the Juvenile Form of Batten Disease

**DOI:** 10.3390/ijms21218055

**Published:** 2020-10-29

**Authors:** Elena K. Shematorova, George V. Shpakovski

**Affiliations:** 1Laboratory of Mechanisms of Gene Expression, Shemyakin-Ovchinnikov Institute of Bioorganic Chemistry, Russian Academy of Sciences, Miklukho-Maklaya 16/10, 117997 Moscow, Russia; elenashe@mail.ru; 2National Research Center “Kurchatov Institute”, 1, Academika Kurchatova pl., 123182 Moscow, Russia

**Keywords:** neuronal ceroid lipofuscinoses, juvenile Batten disease, JNCL, CLN3, *CLN3* gene regulation, biomarker *POLR2J2*, molecular mechanisms

## Abstract

The neuronal ceroid lipofuscinoses (NCLs) collectively constitute one of the most common forms of inherited childhood-onset neurodegenerative disorders. They form a heterogeneous group of incurable lysosomal storage diseases that lead to blindness, motor deterioration, epilepsy, and dementia. Traditionally the NCL diseases were classified according to the age of disease onset (infantile, late-infantile, juvenile, and adult forms), with at least 13 different NCL varieties having been described at present. The current review focuses on classic juvenile NCL (JNCL) or the so-called Batten (Batten-Spielmeyer-Vogt; Spielmeyer-Sjogren) disease, which represents the most common and the most studied form of NCL, and is caused by mutations in the *CLN3* gene located on human chromosome 16. Most JNCL patients carry the same 1.02-kb deletion in this gene, encoding an unusual transmembrane protein, CLN3, or battenin. Accordingly, the names *CLN3*-related neuronal ceroid lipofuscinosis or *CLN3*-disease sometimes have been used for this malady. Despite excessive in vitro and in vivo studies, the precise functions of the CLN3 protein and the JNCL disease mechanisms remain elusive and are the main subject of this review. Although the *CLN3* gene is highly conserved in evolution of all mammalian species, detailed analysis of recent genomic and transcriptomic data indicates the presence of human-specific features of its expression, which are also under discussion. The main recorded to date changes in cell metabolism, to some extent contributing to the emergence and progression of JNCL disease, and human-specific molecular features of *CLN3* gene expression are summarized and critically discussed with an emphasis on the possible molecular mechanisms of the malady appearance and progression.

## 1. Introduction

The juvenile neuronal ceroid lipofuscinosis (JNCL) or Batten disease is a devastating and fatal disorder that is connected to a mutation of the *CLN3* (ceroid lipofuscinosis neuronal 3) gene. Up to 90% of patients with Batten disease carry the 1.02 kb deletion, which, in homozygous form, always causes a severe phenotype, including blindness, epilepsy, dementia, and early death at approximately 20–30 years of age [1]. There are a couple of excellent recent reviews on Batten disease, describing genetic basis [2] and molecular factors of the malady [3,4,5], scrupulously detailing the symptoms of the disease and its diagnosis [1] or summarizing data on what cellular processes are affected by *CLN3* dysfunction [5], but a molecular explanation of how this well-characterized genetic defect (*CLN3*-mutations) affect neuronal physiology is still missing and the function(s) of the CLN3 protein is still elusive. As a rule (in more than 80% of patients), the disease begins with a distinct functional visual impairment and progressive vision loss at the age of 5–10 years, with complete blindness occurs at a later stage of the illness progression [6]. Fine granularity of the retinal pigment epithelium in the central macula and even bull’s-eye maculopathy may be present in the early stages of the disease.

When vision is affected, symptoms such as night blindness, tunnel vision (without peripheral vision) or vice versa—peripheral vision (without central vision), reticular vision, antipathy to bright light, slow adaptation from darkness to light and vice versa, blurred outlines of visible objects—i.e. blurred vision, poor color discrimination—come to the fore [7]. These features are explained by the relative preservation of the upper peripheral retina with the loss of the central and lower fields of vision and are usually manifested by a significantly reduced amplitude of the rod- and cone-response [8]. In the future, Batten disease develops atrophy of the optic nerve, weakening of blood vessels, and accumulation of pigment in the peripheral retina. Some JNCL patients were diagnosed with cataracts and even developed acute glaucoma in the later stages of the disease [9]. In general, the rate of progression of vision loss is extremely fast compared to retinal degeneration in other ophthalmic diseases and leads to legal blindness just a few years after presentation [7].

Progressive vision loss in previously healthy children is accompanied by personality changes, behavioral problems, and slow learning. Psychoneurological manifestations include a gradual loss of social adaptation with frequent and abrupt mood changes, calmness, clumsiness, stumbling, and lethargy [1]. These symptoms are replaced within 2–4 years by even more obvious psychomotor degeneration, leading to convulsive seizures and a persistent decrease in cognitive activity, followed by the loss of acquired skills (including walking, sitting, simple and necessary elements of self-service), loss of speech skills due to damage to the central speech motor apparatus (with understanding of the heard speech remains) [10,11]. This progressive loss of motor and learning (movement and speech) functions begins with Parkinson-like symptoms and leads to dementia. Eventually, those who suffer become disabled, bedridden, and die prematurely.

Batten disease is often difficult to diagnose because it is quite rare. The incidence of *CLN3*-disease in the United States and several European countries (for example, Germany) is approximately 1 per 100,000 live newborns [12]. The highest prevalence of JNCL is found in the Scandinavian countries, especially in Finland (slightly more than 1 in 25,000 live newborns with a carrier frequency of 1 in 70). Enrichment of certain alleles indicated that the same major mutation is responsible for Batten disease in Finland as in most other European countries. Moreover, haplotype analysis suggested that most *CLN3*-deficient chromosomes arose from a single founder mutation [13].

As noted above, vision problems are often the first symptoms. Therefore, the initial diagnosis can be made by an eye examination using the electrophysiological methods mentioned above. To confirm the diagnosis, blood and urine tests are performed to search for lipopigment buildup (e.g., searching for lymphocyte vacuoles in blood by light microscopy), as well as a skin biopsy with its examination under an electron microscope.

The next important step in confirming a diagnosis is to detect a mutation/deletion in the *CLN3* gene. Munroe et al. used PCR to identify the intragenic microsatellite marker D16S298 to make the prenatal diagnosis of Batten disease on the basis of a chorionic villus sample [14]. A rapid diagnostic solid-phase minisequencing test to detect the most common 1.02-kb *CLN3* deletion was developed in Finland [15]. Based on modern direct Sanger sequencing technique, more comprehensive diagnostic tests on all neuronal ceroid lipofuscinoses (NCLs) were recently developed [16]. Later manifested brain abnormalities are diagnosed using MRI scans, computed tomography and electroencephalogram (EEG) measurements.

Nearly half of JNCL patients have abnormalities in other organ and tissues. This includes, for example such cardio deficiencies as T-wave inversion, sinus node dysfunction and ventricular hypertrophy leading to severe bradycardia and/or other conduction abnormalities [17]. The skeletal muscles of all CLN3 patients show varying degrees of atrophy. They are homogeneously affected by an autophagic vacuolar myopathy (AVM) characterized by autophagic vacuoles and characteristic lysosomal pathology [18]. However, it is worth emphasizing that although the *CLN3* gene is expressed everywhere, only certain tissues and organs of human body are affected, and neurological functions of the brain are the most severely damaged.

In this review, an attempt is made to examine all aspects of cellular physiology, which are disturbed in cells containing defective CLN3 protein and contribute to the development of pathological changes in the central nervous system, leading to the emergence and progression of still incurable Batten disease. The important human-specific aspect of expression and regulation of *CLN3* gene is also considered.

## 2. Human-Specific Aspects of the *CLN3* Gene Structure and Expression Which Could Have Influence on the Disease Progression

It was clearly established that juvenile neuronal ceroid lipofuscinosis (JNCL) or Batten disease (CLN3; OMIM #204200) is caused by homozygous or compound heterozygous mutation in the *CLN3* gene (NCBI ID 1201, HGNC ID 2074, MIM ID 607042) located on the short arm of the *Homo sapiens* chromosome 16 at position 12.1 (16p12) [5]. In close vicinity to the *CLN3* in the human genome, there are *APOBR* and *IL27* genes encoding the apolipoprotein B receptor and interleukin 27 (on the right) and gene *NPIPB7* of nuclear pore complex interacting protein family member B7 (even with partial overlap with *CLN3*) and still uncharacterized locus *LOC112268171* (on the left).The mutation responsible for about 75% of JNCL disease is a genomic deletion of 1.02 kb, including 217 nt of the open reading frame (nucleotides 598–814) corresponding to two coding exons (number 7 and number 8) [19]. Elimination of these 217 bp of coding sequence produces a frameshift, generating a TAA termination codon 84 bp downstream of the deletion junction. The predicted translation product is a truncated protein of 181 amino acids consisting of the first 153 residues of the CLN3 protein, followed by 28 novel amino acids before the stop codon. Munroe et al. identified homozygosity for this common 1.02 kb *CLN3* deletion in 139 (74%) of 188 unrelated patients with Batten disease [14]. The fact that most patients appear to be homozygous for a common *CLN3* deletion fits with the prediction of *CLN3* as not being haploinsufficient [20]. Mole and Cotman tabulated the mutations that had been identified in the various CLNs; they reported 67 mutations, 2 promoter changes and 13 polymorphisms associated with *CLN3* [2]. The 181 amino acids long truncated protein produced from the common 1.02 kb deletion mutant allele retains significant CLN3 function, indicating that JNCL is a mutation-specific disease phenotype [21]. The residual function likely explains why classic Batten disease shows later onset and less severe clinical manifestations compared to other forms of neuronal ceroid lipofuscinoses (NLCs).

The full-length CLN3 protein, also called battenin, contains 438-amino acids. The amino acid sequence of the *CLN3* gene product predicts a hydrophobic protein with 6 transmembrane domains [22]. The sequence appears to contain a putative mitochondrial targeting site at residue 11 and has four predicted *N*-glycosylation sites, all on the predicted exoplasmic surface. The CLN3 protein sequence does not display significant similarities to any protein of known function.

Although the *CLN3* gene is highly conserved in evolution (from yeast to humans, and especially in mammals), genomic studies show that there are human-specific traits in its expression mechanisms. Indeed, as some other adaptively evolving genes [23], human *CLN3* gene has two different transcription start points probably with very different regulation. The identical full-length hCLN3 protein can be synthesized from two different mRNAs transcribed from the same gene by using specific and alternatively regulated promoters (Figure 1).

Moreover, it is very likely that one of the transcripts (isoform 1) is generated by a specific form of RNA polymerase II containing minor isoform of the subunit hRPB11 (POLR2J), the product of expression of a human-specific *POLR2J2* gene [24,25,26,27,28]. Recent identification of this (*POLR2J2*) gene as one of the only five identified biomarker genes for Batten disease [29] strongly argues in favor of this possibility.

To identify genetic variations responsible for the clinical variability of Batten disease, Lebrun et al. performed genome-wide microarray analyses in lymphocytes derived from patients with three different speed of the disease progression (slow, average and rapid). Five genes, called biomarkers, were found to be dysregulated by a similar way in all three patient groups: the expression of *POLR2J2* and *CDC42SE2* were decreased, whereas *RGS1*, *DUSP2,* and *PARP15* were increased, in all CLN3 patients compared with controls.

The human specificity of the *POLR2J2* and *POLR2J3* genes was proven [24,25,27] and participation of products of their expression, minor subunits hRPB11bα (hRPB11cα) and hRPB11bβ (hRPB11cβ) of human RNA polymerase II, in formation of novel, human-specific transcription complexes was demonstrated in our previous works [28,30,31].

One of the surprising facts is that so far no large animals (dogs, cows, goats, etc.) have been found to suffer from a malady similar in symptoms to Batten disease (i.e., from the mutations or deletions in the *CLN3* gene) [32]. To search for such animals, a separate project was created, which made it possible to detect large animals with mutations in almost all other *CLN*-genes discovered at that time [33,34,35]. In 2019, pigs were created that had a deletion in their *CLN3* gene similar to that of patients with Batten disease (JNCL) [36]. However, at the moment there is still no information about what symptoms these animals show. For today the only mammalian species where the function of the *CLN3* gene has been systematically addressed is mouse. Like its human counterpart, the mouse cDNA encodes a predicted 438-amino acid polypeptide which share 85% sequence identity with human protein [37]. Despite the obvious membrane deposition, *Cln3*^−/−^ knock-in mice did not develop obvious clinical symptoms by 12 months of age [38]. And only in the second half of life, symptoms appear in a mild form: visual impairment, neuron loss, motor disfunctions, reduction of lifespan by 20% [39,40].

The presence of human-specific aspects of *hCLN3* gene expression seems also related to the characteristic features of human neurons. In mice, monkeys, and other animals, the size of a neuron is limited by the level of expression of membrane proteins. At the same time, human cortical neurons are significantly larger and have a much larger number of dendrites. Moreover, in humans compared to all studied animals there is a drastic change in dendritic length with increasing radical distance from cell body [41]. In such specific cells as neurons, the limiting stage of their growth is the formation of giant membrane surfaces, and the CLN3 protein is undoubtedly involved in this process (discussed in [42]). It is quite possible that children in the period of active growth of their neurons need very effective production (high-performance synthesis) of battenin (CLN3 protein), which is achieved by the increased expression of the *CLN3* mRNA isoform 1 (Figure 1), using a special form of RNA polymerase II containing the human-specific subunit POLR2J2 (hRPB11bα) in its composition (see Refs [28,30,31] for a discussion).

The isoform 1 *CLN3* mRNA so far was predominantly isolated from the nervous tissue and thus may be at least to some extent brain-specific. In humans (at some stage of their development) this isoform is probably transcribed by the human-specific form of RNA polymerase II containing minor subunit POLR2J2 (hRPB11bα or hRPBbβ) [25,28,31].

Although originally, Batten disease was classified as one of the lysosomal storage disorders, to find the true causes of the disease, all other aspects of changes in cellular physiology need to be considered. Moreover, it is important to find a molecular explanation of how the well-characterized genetic defect (disruption of the *CLN3* gene function) could affect cellular physiology of the neuronal tissue in a first place—the knowledge that is still largely missing. In the following sections of the review, we will sequentially consider specific physiological changes in *cln3^−^* cells (both from humans and from different model organisms, such as yeasts, slime mold, fruit fly and mouse) and discuss cellular systems and functions, primarily affected or damaged during progression of this neurodegenerative disease (Table 1).

## 3. The CLN3 Protein Mediates Anterograde and Retrograde Trafficking of Proteins and Other Substances

Production and distribution (delivery to destination) of СLN3 membrane protein is a rather complex and finely regulated process: СLN3 is synthesized in the endoplasmic reticulum, then transported to the Golgi apparatus and sent to the membranes of many cellular organelles (mitochondria, early and late endosomes and lysosomes) [43]. Since СLN3 is a component of the endosomal-lysosomal system, endocytosis [44], amino acid transport in vacuoles [45], autophagosomal maturation, as well as the stage of endosomal and lysosomal fusion are disturbed in the cells of JCNL patents [46]. Defects in the lysosomal-endosomal system lead to the formation of lysosomal inclusions, which have long been considered the cause of neurodegenerative processes occurring in patients suffering from Batten disease.

To identify components critical to lysosomal homeostasis that are affected by Batten disease, lysosomal proteomes were established in cerebellar cell lines derived from a *Cln3*-knock-in mouse model of human Batten disease and control cells. This analysis identified 70 proteins assigned to the lysosomal compartment and 3 lysosomal cargo receptors, which in most cases showed significant differential abundance between control and *CLN3*-defective cells. Among them, 28 soluble lysosomal proteins catalyzing the degradation of various macromolecules had decreased levels in *CLN3*-deficient cells. A decrease in the activity of 11 lipid-degrading lysosomal enzymes correlated with a reduced capability to degrade lipid droplets and some changes in the distribution and composition of membrane lipids. In particular, the levels of glycosphingolipids and lactosylceramides were decreased in *CLN3*-defective cells, which were also impaired in the exocyte transferrin receptor recirculation pathway [47]. Another signaling hub which includes two major mitogen-activated protein kinase (MAPK) cascades, and centers on the Tor kinase complexes TORC1 and TORC2, was identified in a global screen for genetic interactions with *btn1*, the fission yeast *Schizosaccharomyces pombe* orthologue of a human *CLN3* gene [48]. In particular, it was shown that the TORC signaling pathway is repressed in yeast cells with deletion of the *btn1* gene, which can also have a significant impact on the effectiveness of autophagy and lysosomogenesis.

Because Batten disease is primarily a neurodegenerative disorder, an important issue is the function of CLN3 membrane protein in brain tissue. In this regard, it is worth noting the work of Luiro and colleagues [49] who demonstrated that СLN3 is found in synaptosomal fractions isolated from nerve endings. This may involve interaction of CLN3 at the plasma membrane with components that regulate the actin cytoskeleton (β-fodrin and non-muscle myosin-IIB) [50,51] and/or with lipid rafts that are enriched in cholesterol and galactosylceramide [52,53]. The endocytic defect caused by *CLN3* mutations can change neurotransmitter release and surface expression of receptors [49] and ion channels in neurons [72], leading to impaired neurotransmission and eventually neuronal cell death (Figure 2).

## 4. Stress of the Endoplasmic Reticulum (ER Stress)

Although trafficking of proteins is a pathway connecting the Golgi network, endosomes, autophagosomes, lysosomes and plasma membrane, it begins in the endoplasmic reticulum where the primary sorting of proteins occurs. A common feature of many neurodegenerative diseases is the accumulation of misfolded proteins that cause ER stress, affecting neuronal homeostasis and leading to cell death [54,73]. During times of ER stress, transcription of genes encoding secretory proteins is downregulated, misfolded proteins are cleared through ER-associated degradation (ERAD), more chaperones are synthesized in an attempt to increase the protein folding capacity. If the aforementioned mechanisms fail to rescue and restore cell homeostasis, then apoptosis could be triggered through the intrinsic pathway in an attempt to prevent further damage at the level of tissue or organism.

Eight NCL proteins can be connected in a single protein network [42]. Their interactions can occur only in the endoplasmic reticulum, so these proteins might have some ER-associated functions. Recently, it was shown that CLN8 is an endoplasmic reticulum cargo receptor [74] and CLN7 influences the secretion of Cln5 and CtsD in *Dictyostelium discoideum* [75].

To study the function of CLN3 in the ER-stress signaling pathway, Wu et al. compared proliferation and apoptosis in cells transfected by normal and mutant *CLN3* after induction of ER stress with tunicamycin (TM). This drug blocks enzyme GlcNAc phosphotransferase (GPT), which catalyzes the transfer of *N*-acetylglucosamine-1-phosphate from UDP-*N*-acetylglucosamine to dolichol phosphate in the first step of glycoprotein synthesis. CLN3 protected cells from TM-induced apoptosis and increased cell proliferation. Increased expression of wild type *CLN3* gene resulted in increased production of the ER chaperone protein GRP78 and decreased level of synthesis of proapoptotic protein CHOP. In contrast, increased expression of mutant *cln3* or knockdown of *CLN3* gene by siRNA had the opposite effect [55]. These findings suggest that СLN3 might be some kind of ER chaperone involved in the folding of glycoproteins, and its dysfunction in cells could lead to failure of the ER stress response program with switching on mechanism of apoptosis. That is why ER stress may be a key mechanism in the development of JNCL disease causing neuronal degeneration.

## 5. Dysfunction of Mitochondria

Mitochondrial dysfunction may also be one of the causes of neurodegeneration [56]. Morphological changes of mitochondria were observed in patients with JNCL: they are enlarged and lengthened [57]. Studies of skin fibroblasts from JNCL patients revealed a decrease in the basal activity of ATP synthase [58], the overall level of oxidative processes, a decrease in the concentration of AMP and energy-rich phosphates ATP and ADP [59]. In the mitochondria of JNCL patients, the rate of synthesis of fatty acids decreases, which leads to a mitochondrial membrane defect [60]. In addition, ATP synthase complexes function abnormally, since a large number of them are destroyed, and one of the subunits of this enzyme (namely, *C*) accumulates in lysosomal inclusions, which is one of the distinctive signs of Batten disease [61].

It was also shown that JNCL and other neuronal ceroid lipofuscinoses are characterized by selective loss of inhibitory GABAergic interneurons rich in mitochondria and are thus especially susceptible to compromised energy production [49]. Neuron loss primarily in brain areas most metabolically active and rich in mitochondria (cortical layers IV–V) in the JNCL patients also provides indirect evidence of mitochondrial involvement in the JNCL pathogenesis [62]. It was observed that genes of glycolysis (*Gpi*, *Eno1*, *Tpi* and *Pfk1*) are upregulated in embryonic primary culture of the *Cln3*^−/−^ mouse neurons. Pfk1 (phosphofructokinase 1) is specific only to glycolysis, suggesting that upregulation of glycolysis may act as a compensatory mechanism for an inefficient energy production. Analysis of the mitochondrial function of the aforementioned *Cln3*-deficient mice also shows slightly reduced activities of the mitochondrial respiratory chain complexes, indicating reduced utilization of oxygen [49].

The negative effect on mitochondrial function in *cln3^−^* cells may be due to the suppression of the *TORC1* gene function and the repression of TOR signaling, as observed in the cells of the fission yeast *Schizosaccharomyces pombe* with the deletion of the *btn1* gene, a homologue of human *CLN3* [48].

## 6. Reactive Oxygen Species (ROS)

An increase of reactive oxygen species (ROS) mediates different pathological processes such as atherosclerosis, diabetes, neurodegeneration, inflammation, and aging [76]. The observed alterations in macroautophagy can lead to an accumulation of oxidatively damaged mitochondria that in turn fuel the generation of ROS. Excess cellular levels of ROS cause damage to proteins, nucleic acids, lipids, membranes and organelles, which can lead to activation of cell death processes such as apoptosis. It was shown that ROS were significantly increased in *cln3* deficient Batten disease patients fibroblasts in comparison with their appropriate controls [63]. Experiments with *Drosophila melanogaster* as a relatively simple model organism for studying Batten disease showed that the fruit fly cells lacking *CLN3* are hypersensitive to oxidative stress yet they respond normally to other physiological stresses. Overexpression of the *CLN3* gene is sufficient to confer increased resistance to oxidative stress. These data suggest that the lack of CLN3 function leads to a failure to manage the response to oxidative stress and this may be the key deficit in JNCL that leads to neuronal degeneration [64]. An increase of reactive oxygen species (ROS) can modify the cell-signaling proteins and have functional consequences, which mediate pathological processes neurodegeneration. As it was previously shown and reviewed, the generation and homeostasis of intracellular ROS could have impact on various cell-signaling (NF-*κ*B, MAPKs, PI3K-Akt) and Ubiquitination/Proteasome systems, modifying protein kinases and ion channels and transporters (Ca^2+^ and mPTP) [77].

## 7. pH Homeostasis and Osmoregulation

Previously, it has shown that the *BTN1* gene product of the yeast *Saccharomyces cerevisiae* is 39% identical and 59% similar to human CLN3, which is associated with the neurodegenerative disorder Batten disease. Yeast *btn1*-Δ strains have an elevated activity of the plasma membrane H^+^-ATPase due to an abnormally high vacuolar acidity during the early phase of growth [65].

Interesting results have been obtained with the social amoeba *Dictyostelium discoideum*, a soil microbe that has long been used as a model system for studying fundamental processes in cell biology and human neurological disorders, including NCLs. During starvation, *Dictyostelium* cells move from the stage of unicellular feeding, when cells grow and divide mitotically, to the stage of multicellular development. During development, individual cells aggregate and undergo a process of differentiation, forming a fruiting body, consisting of a stem of cells that support a mass of viable spores. In *Dictyostelium discoideum*, the CLN3 protein is localized primarily in the contractile vacuole (CV) system, which consists of a network of tubules that supply water to the bladders. During hypotonic stress, these bubbles of a social amoeba fill with water, and then periodically displace the accumulated water from the cell, collapsing on the plasma membrane. This mechanism allows cells to effectively regulate osmotic pressure in a variety of environments. It was shown that *cln3*^−^ cells are sensitive to both hypotonic and hypertensive stress, which ultimately affects their viability and ability to multicellular development [66].

Hypertensive stress increased the expression of the *CLN3* gene in the kidney cells of young hamsters and influenced the localization of the protein [67]. In addition, work on mice suggested a role for CLN3 in renal control of water and/or ionic homeostasis [68]. Finally, direct evidence for a role for CLN3 in osmoregulation was found when it was shown that regulatory volume depletion (RVD) is impaired in brain endothelial cells derived from *CLN3*-deficient mice [69]. RVD is a normal process that occurs when cells live in a hypotonic environment. The cells grow in size by absorbing excess water and then return to their normal size, displacing the water. Thus, if the process of osmoregulation is disturbed, the cell can be destroyed due to the inability to throw out excess water.

## 8. The Hyperactivation of Glial Cells and Astrocytes

Microglia are the resident immune cells within the central nervous system (CNS) parenchyma. The main functions of microglia are to phagocytose dead cells and release pro-inflammatory mediators. Abnormal neuron-microglia interactions have been implicated in the pathogenesis of several neurodegenerative diseases, including Alzheimer’s and Parkinson’s disease [78]. Activated microglia are also observed in the brains of JNCL patients [70]. It was reported that microglia cells carrying the most common deletion of *CLN3* gene (*CLN3*Δ ex7/8) exist in a pro-inflammatory state. Upon stimulation, microglia produce significantly more mediators of inflammation, including IL-1β, TNF-a, IL-1a, IL-9, and IL-15 [71]. *cln3*-deficient astrocytes are characterized by the reduced secretion of a range of neuroprotective factors, mitogens, chemokines and cytokines, in addition to impaired calcium signaling and glutamate clearance. Using a co-culture system, it was shown that *cln3*-deficient astrocytes and microglia had a negative impact on the survival and morphology of both *cln3*-deficient and wild type neurons. These data provide evidence that astrocytes in time of CLN3 disease are functionally compromised. Together with microglia, they may play an active role in neuron loss in this disorder and can be considered as potential targets for therapeutic interventions [71].

## 9. Unregulated Activation of Signal Cascades

In Shematorova et al. [42], it was suggested that СLN3 is a chaperone of the endoplasmic reticulum and is involved in the formation of the three-dimensional structure of certain receptors, and its absence leads to changes in the 3D structure of these type of proteins. Incorrect folding of the plasma membrane proteins leads to initiation of signaling cascades from the cell surface. Although the true receptors that can trigger a signaling cascade from the membrane surface were not shown in the work, Uusi-Rauva et al. revealed that CLN3 interacts with ion channels such as Na^+^/K^+^ ATPase ATP1A1 (α1) subunit [50]. The α1 (first catalytic) subunit of Na^+^/K^+^ ATPase possesses both pumping and signaling functions. This protein, ATP1A1, might interact with non-receptor tyrosine kinase SRC and regulates its activity in a conformation-dependent manner [79]. It was also shown that cardiac glycoside ouabain [79], hydrogen peroxide [80], and ammonium chloride [81] might induce conformational changes in the tertiary structure of Na^+^/K^+^ ATPase ATP1A1 that leads to SRC and subsequent EGFR activation. So, conformational change in α1 Na^+^/K^+^ ATPase and its activation of SRC can be easily transmitted to MAPK cascades.

Indeed, recent studies have shown that SRC activation is required for ouabain-induced activation of ERK1/ERK2 and p38 [82]. MAPK (mitosis activated protein kinases) ERK1/ERK2 are responsible for cell growth and proliferation and are involved in the activation of genes that provide synthesis of growth factors and mitogens. However, excessive uncontrolled activation of these signaling pathways can lead to the overproduction of certain proteins and their toxicity, which can lead to cell death. The p38 and JUN pathways have also been implicated in stress-induced signaling leading to apoptosis [83]. Thus, CLN3 may be involved in the correct formation of the spatial structure of ATP1A1 in neurons. Mutations or truncations of this protein lead to a shift in ATP1A1 signaling and activation of EGFR via SRC kinase [83] (Figure 3).

Recently by using comparative transcriptomics Huber and Mathavarajah identified genes whose expression is significantly increased in the social amoeba *Dictyostelium discoideum cln3**^−^* cells during starvation. In these conditions *Dictyostelium* cells secrete cAMP, which acts as a chemoattractant causing cells to aggregate into multicellular mounds and after that in the multicellular fruitful body, forming spores [84]. *cln3**^−^*cells displayed delayed migration and aggregation that support aberrant signaling during this stage of development [83]. Expression of a number of genes encoding MAPK signaling cascade proteins are elevated under starvation conditions in *Dictyostelium*
*cln3**^−^* cells, which indicates the importance of CLN3 for proper functioning of this particular regulatory pathway [85].

## 10. Current Treatment Opportunities

Perhaps the most promising method of treating JNCL today is the introduction of the normal *CLN3* gene into the spinal cord of patients using adeno-associated virus (AAV) constructs [86,87]. Indeed, AAV-mediated gene therapies have proven as a successful treatment strategy to combat photoreceptor degenerations. Kleine Holthaus et al. developed a successful treatment for the retinal degeneration in *Cln6*-deficient mice, a model of NCL that carries a mutation in the *Cln6* gene [88]. When the therapeutic *CLN6* transgene was delivered specifically to bipolar cells, a retinal cell type downstream of photoreceptors, degeneration of photoreceptor cells became significantly less strong [89].

Another important breakthrough has been the application of enzyme replacement technique (Bruniera) for CLN2 disease, the first ever FDA-approved therapy, which is significantly improving quality of life and delaying disease progression for persons affected by this disease [90]. Unfortunately, this type of “cross-correction” which is applicable for enzyme deficient forms of NCL could not be used in case of Batten disease.

At the moment, phase I/IIa clinical trials of gene therapy are currently in progress for two other forms of NCL (CLN3 disease: NCT0377057233 and CLN6 disease: NCT0272558034). In case of CLN3 (Batten) disease, recently initiated by Amicus Therapeutics treatment consists of a one-time low or high dose injection of AT-GTX-502 construct (scAAV9.P546.CLN3) into the lumbar spinal cord region of subjects with CLN3 Batten disease and their monitoring over a period of at least three years [91].

Concerning the examples of gene therapy approaches mentioned above, the main issues are the presence of naturally occurring antibodies against adeno-associated viruses (AAVs) in a variable proportion of the patient population and the open question whether transgene expression will be sustained for a long period of time — re-administration of vectors right now is hardly possible [92].

Another way of treatment is to use a splice-modulating antisense oligonucleotides drug tailored to a particular patient [93]. There is no serious adverse events, and treatment leads to objective reduction in seizures (determined by electroencephalography and parental reporting). However, this method can only be used for patients with proven aberrant splicing in the synthesis of *CLN3* mRNA, as a result of which the correct reading frame of the battenin (CLN3 protein) cannot be formed.

Using mouse models, drugs have been discovered that fight certain cell disorders caused by CLN3 deficiency, such as neuroinflammation [94,95], imposed autophagy and reduced mitochondrial membrane potential [96], apoptosis [97]. Mycophenolate, an immunosuppressant, today is already used in the treatment of patients with Batten disease [98].

Unfortunately, all the methods listed above are universal for the treatment of all monogenic recessive hereditary diseases and do not rely on understanding the mechanisms of development of a particular disease.

## 11. Conclusions

Batten disease or JNCL is an incurable devastating disease affecting primarily young children. Currently, there is no curative therapy for this malady, and all available treatments for JNCL are symptomatic and palliative. Although originally Batten disease was classified as one of the lysosomal storage diseases and many studies were aimed at investigating this particular aspect of the malady, these studies did not lead to understanding of the molecular mechanisms of this disease and/or finding potential ways to treat it. The main mechanisms of Batten disease as a neurodegenerative disorder can be associated with many other defects in cellular physiology summarized and discussed in the present review.

As noted earlier, all animal models of JNCL disease studied so far do not exhibit human-specific symptoms. For example, the introduction of 1.02-kb deletion in the mouse *cln3* gene causes the retina degeneration only in the second half of the animals’ life, and all the other symptoms are manifested in a mild form [32]. Therefore, it is not surprising that at the moment only palliative care is available for patients with Batten disease. It is possible that only gene therapy using adenovirus-based vectors can bring some positive effect(s), as was observed in the case of enzyme replacement therapy for Jansky-Bielschowsky disease (NCL2) caused by mutations in the *CLN2* gene [90].

It is likely that the disease process may involve a special battenin-dependent pathway that is essential for neuronal cell survival, and thus consistent with battenin-specific function in neurons. In connection with the discovery of human-specific features of *CLN3* gene expression, it is important to study in detail both promoters of this gene and the expression of two isoforms of its full-size mRNA in different tissues, as well as the possible participation of human-specific genes *POLR2J2* (and *POLR2J3*) in this process. This may help, in particular, to clarify the interesting fact noted by some researchers that although the level of *CLN3* mRNA in the brain is low compared to other tissues (please, see Transcriptome Databases), it is in the cells of the nervous tissue that the extremely high (the highest) level of synthesis of the battenin (CLN3 protein) is observed [99]. A thorough understanding of the mechanisms regulating disease progression can help determine new treatments for Batten disease. Without an understanding of the molecular mechanisms of Batten disease, it will be impossible to fight the illness.

## Authors Contributions

E.K.S. conceived the work, designed and drafted the manuscript; G.V.S. conceived the work, revised the manuscript and supervised the work. Both authors have read and approved the final manuscript. All authors have read and agreed to the published version of the manuscript.

## Figures and Tables

**Figure 1 ijms-21-08055-f001:**
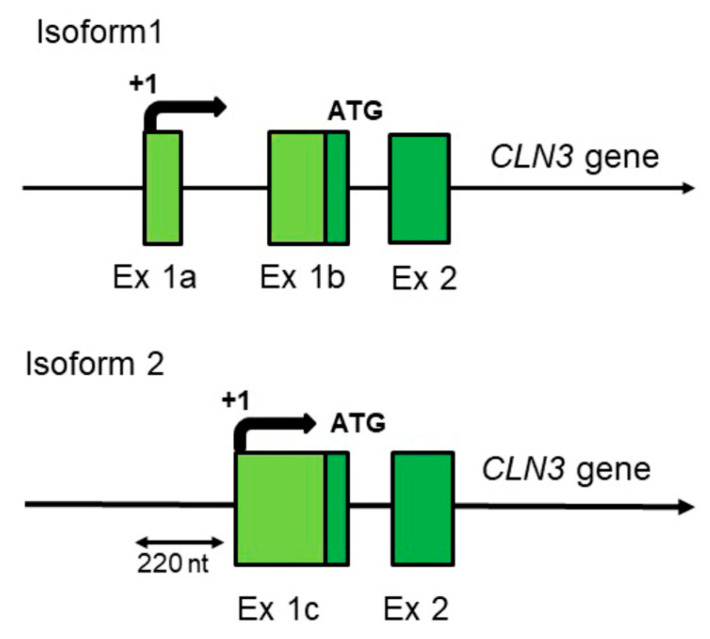
The schematic structure of two different full-length (complete) *CLN3* mRNA isoforms transcribed from varying promoters of the *CLN3* gene of *Homo sapiens*. Only regions corresponding to distinguished (differing) transcription start points and 2–3 of the first exons (out of 15–17) of the *hCLN3* gene are shown. *hCLN3* isoform 1 (transcript variant 1)—3832 nt (GenBank NM_001042432, UniProtKB/Swiss-Prot Q13286, CCDS 10632, Ensembl-Tr: ENST00000636147 (transcript: CLN3-244), protein—ENSP00000490105); hCLN3 isoform 2 (transcript variant 2)—1876 nt (GenBank NM_000086, UniProtKB/Swiss-Prot Q13286, CCDS 10632, Ensembl-Tr: ENST00000359984 (transcript: CLN3-205), protein—ENSP00000353073).

**Figure 2 ijms-21-08055-f002:**
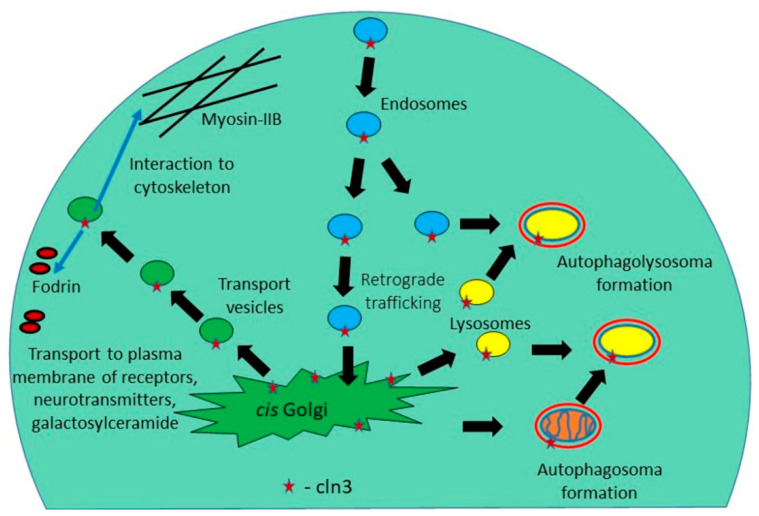
The CLN3 protein mediates anterograde and retrograde trafficking of Golgi network—plasma membrane, participates in the formation of endosomes, autophagosomes, and lysosomes.

**Figure 3 ijms-21-08055-f003:**
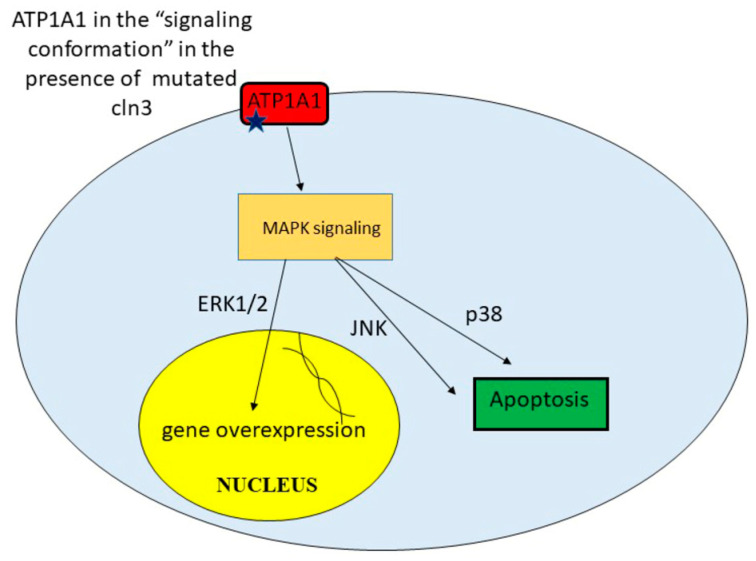
Processes activating in *cln3*^−^ cells in presence of unregulated signaling cascades.

**Table 1 ijms-21-08055-t001:** Possible molecular mechanisms of the JNCL (juvenile neuronal ceroid lipofuscinosis).

Physiological Changes in *cln3^−^* Cells and/or the Cellular System Affected	Probable Cause of the Cell Death	References
Endosomal-lysosomal system, autophagy	Defects in traffic of proteins, cholesterol, sphingolipids, neurotransmitters. Accumulation of damaged organelles	[43,44,45,46,47,48,49,50,51,52,53]
Stress of endoplasmic reticulum (ER stress)	Accumulation of misfolded proteins and, as a consequence, apoptosis	[54,55]
Dysfunction of mitochondria	Deficiency in energy-rich phosphates ATP and ADP	[56,57,58,59,60,61,62]
Reactive oxygen species (ROS)	Damage of proteins, nucleic acids, lipids, membranes and organelles, as a consequence — apoptosis	[63,64]
pH homeostasis and osmoregulation	Сell destruction due to the inability to throw out excess of water	[65,66,67,68,69]
The hyperactivation of glial cells and astrocytes	Stimulation of inflammation	[70,71]
Upregulated activation of signal cascades	Excessive amount of some proteins leads to toxicity. Apoptosis	[42]

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
