# Peer review of "Current Insights in Elucidation of Possible Molecular Mechanisms of the Juvenile Form of Batten Disease"

_ijms, 2020, doi:10.3390/ijms21218055_

Round 1
Reviewer 1 Report
This review focuses on JNCL the most common form of Batten disease which is caused by mutations in the CLN3 gene. The first part of the review talks about the most common deletion and how specific isoforms of the gene are present in humans due to differential transcription. The review then focuses on some of the known alterations to pathways and systems in JNCL discovered in different models ranging from yeast to mice to human cell systems and how these disruptions may explain the physiology seen in human patients. I have some comments as seen below.
I find that the two sections are not connected quite clearly enough and this could perhaps be resolved with some more explanatory sentences between the two sections.
The paragraph (Lines 122-128) has 2 sentences regarding similarity of human protein to mouse protein. It seems out of place here given the section is specific to humans. The discussion of the protein after this sentence is presumably in relation to human protein not mouse but it is not clear which the authors are referring to. I think that if the authors are talking about the human protein then the sentences regarding the mouse similarity are not needed here.
Table 1 could do with some references and the organism this was identified in. Also, in the heading which cells are they referring to in cause of cell death?
Table 1 and the following sections are about CLN3 protein not just human specific isoforms, but that is not clear as it flows directly from talk about human specific physiology and human specific isoforms. Perhaps a sentence here to explain this would help.
Line 273 human fibroblasts? The authors should note which organism the fibroblasts are from.
There is no discussion about TORC1 which would be beneficial especially given the sections on mitochondria, autophagy, amino acids etc.
Author Response
Dear Editors,
We are sending you our manuscript entitled “Current insights in elucidation of possible molecular mechanisms of the juvenile form of Batten disease” which has been carefully revised according to yours and Reviewers' comments. Hope it can now meet all the requirement of the "International Journal of Molecular Sciences”.
The revised text of the article takes into account all critical comments of the Reviewers
1 & 2 which we highly appreciate.
Point by point responses on specific issues raised by the Reviewers 1 & 2 are shown below and also clearly indicated in version of the manuscript with Track Changes provided as a separate file: ijms-966872 (with Track Changes).
Reviewer 1
Reviewer 1:
Comments and Suggestions for Authors
This review focuses on JNCL the most common form of Batten disease which is caused by mutations in the CLN3 gene. The first part of the review talks about the most common deletion and how specific isoforms of the gene are present in humans due to differential transcription. The review then focuses on some of the known alterations to pathways and systems in JNCL discovered in different models ranging from yeast to mice to human cell systems and how these disruptions may explain the physiology seen in human patients. I have some comments as seen below.
I find that the two sections are not connected quite clearly enough and this could perhaps be resolved with some more explanatory sentences between the two sections.
--- As the Reviewer has suggested, we added explanatory sentences between the sections mentioned (just before Table 1): Lines 193-202 in ‘MS with Track Changes’.
The paragraph (Lines 122-128) has 2 sentences regarding similarity of human protein to mouse protein. It seems out of place here given the section is specific to humans. The discussion of the protein after this sentence is presumably in relation to human protein not mouse but it is not clear which the authors are referring to. I think that if the authors are talking about the human protein then the sentences regarding the mouse similarity are not needed here.
--- As the Reviewer has suggested, two sentences (previous Lines 122-128) regarding mouse Cln3 protein, have been eliminated here and were transferred in more appropriate place (Lines 160-163).
Table 1 could do with some references and the organism this was identified in. Also, in the heading which cells are they referring to in cause of cell death?
---All relevant References have been included in new version of the Table 1 (Lines 209-210) which provide the general overview of the problems (physiological changes in different type of cln3- cells) discussed in the following specific sections. The organisms under investigation are clearly indicated in the References provided and in the text of every specific section.
Table 1 and the following sections are about CLN3 protein not just human specific isoforms, but that is not clear as it flows directly from talk about human specific physiology and human specific isoforms. Perhaps a sentence here to explain this would help.
---The appropriate explanatory sentences have been introduced just before Table 1: Lines 193-202 in ‘MS with Track Changes’. The presence of human-specific traits in the regulation of CLN3 gene expression, first noted in this review, provides a good opportunity to rational design of future studies tentatively suggested in Lines 174-178 & 188-191.
Line 273 human fibroblasts? The authors should note which organism the fibroblasts are from.
--- As we indicated now in the text (Lines 328-329), these are fibroblasts of patients suffering from Batten disease. Thank you for pointing this uncertainty!
There is no discussion about TORC1 which would be beneficial especially given the sections on mitochondria, autophagy, amino acids etc.
---The discussion about importance of TORC signaling is added (Lines 244-249, Lines 318-320 and Ref. [48]).
Sincerely yours,
George V. Shpakovski, D.Sc., Ph.D. ([email protected])
(Corresponding author)
Reviewer 2 Report
This is a well-written and timely review on the Batten Syndrome and CLN3. I think it would be of great interest for a number of researchers and clinicians in the field of human genetics and developmental disorders. I have only a few suggestions for how the review could be improved.
-Define the gene acronym CLN3 when it first appears.
-The Introduction is, in my opinion, a bit too detailed, and some of the issues brought up would fit better under their proper place below.
-Rows 108-121: The fact that most patients appear to be homozygous for a common CLN3 deletion fits with the prediction of CLN3 as not being haploinsufficient (gnomAD; PMC7334197). It may be worth pointing this out.
-I felt that the mouse research on Cln3 was not so extensively described, and several publications not referenced (PMIDs 12374761, 17855597, 10440905, 15326100, 27101989, 10527801; for the record, I am not a co-author of any of these studies). Because the mouse it the only mammalian species where the function of this gene has been systematically addressed, I would recommend a separate section on this topic.
-There is an extensive text regarding the cellular role of CLN3, rows 183-391, but no accompanying figures. I think that some figures would be very helpful for the reader.
Author Response
Dear Editors,
We are sending you our manuscript entitled “Current insights in elucidation of possible molecular mechanisms of the juvenile form of Batten disease” which has been carefully revised according to yours and Reviewers' comments. Hope it can now meet all the requirement of the "International Journal of Molecular Sciences”.
The revised text of the article takes into account all critical comments of the Reviewers
1 & 2 which we highly appreciate.
Point by point responses on specific issues raised by the Reviewers 1 & 2 are shown below and also clearly indicated in version of the manuscript with Track Changes provided as a separate file: ijms-966872 (with Track Changes).
Reviewer 2
Reviewer 2:
Comments and Suggestions for Authors
This is a well-written and timely review on the Batten Syndrome and CLN3. I think it would be of great interest for a number of researchers and clinicians in the field of human genetics and developmental disorders. I have only a few suggestions for how the review could be improved.
-Define the gene acronym CLN3 when it first appears.
--- DONE (Line 35).
-The Introduction is, in my opinion, a bit too detailed, and some of the issues brought up would fit better under their proper place below.
---We left the Introduction intact as it deals mostly with Batten disease sympthomatics and diagnosis (from a historic perspective) – the subjects that are not discussed in further sections of the review.
-Rows 108-121: The fact that most patients appear to be homozygous for a common CLN3 deletion fits with the prediction of CLN3 as not being haploinsufficient (gnomAD; PMC7334197). It may be worth pointing this out.
--- DONE (Line 118-119 & Ref. 20). Thank you for a nice suggestion!
-I felt that the mouse research on Cln3 was not so extensively described, and several publications not referenced (PMIDs 12374761, 17855597, 10440905, 15326100, 27101989, 10527801; for the record, I am not a co-author of any of these studies). Because the mouse it the only mammalian species where the function of this gene has been systematically addressed, I would recommend a separate section on this topic.
---We did not devote a separate chapter on research in mouse species, but have substancially increased discussion on the topic in a relevant place of the review (Lines 160-166). Following the Reviewer recommendation, we have added some important mouse research on Cln3 to the Reference list (Refs. 37-40). We must also emphasized that many other works on the mouse Cln3-model are cited and discussed in different section of the text.
-There is an extensive text regarding the cellular role of CLN3, rows 183-391, but no accompanying figures. I think that some figures would be very helpful for the reader.
---Two new Figures (2 & 3) have been added to the manuscript – with compliments to the Reviewer for this suggestion.
Sincerely yours,
George V. Shpakovski, D.Sc., Ph.D. ([email protected])
(Corresponding author)